# Bismuth-Decorated Beta Zeolites Catalysts for Highly Selective Catalytic Oxidation of Cellulose to Biomass-Derived Glycolic Acid

**DOI:** 10.3390/ijerph192316298

**Published:** 2022-12-05

**Authors:** Fenfen Wang, Dongxue Qu, Shaoshuai Wang, Guojun Liu, Qiang Zhao, Jiaxue Hu, Wendi Dong, Yong Huang, Jinjia Xu, Yuhui Chen

**Affiliations:** 1State Key Laboratory of Materials-oriented Chemical Engineering, College of Chemical Engineering, School of Energy Science and Engineering, Nanjing Tech University, Nanjing 211816, China; 2Joint International Research Laboratory of Biomass Energy and Materials, Jiangsu Co-Innovation Center of Efficient Processing and Utilization of Forest Resources, College of Materials Science and Engineering, Nanjing Forestry University, Nanjing 210037, China; 3Department of Chemistry and Biochemistry, University of Missouri St. Louis, One University Boulevard, St. Louis, MO 63121, USA

**Keywords:** biomass, cellulose, Bi/β, hydrolysis oxidation, glycolic acid

## Abstract

Catalytic conversion of cellulose to liquid fuel and highly valuable platform chemicals remains a critical and challenging process. Here, bismuth-decorated β zeolite catalysts (Bi/β) were exploited for highly efficient hydrolysis and selective oxidation of cellulose to biomass-derived glycolic acid in an O_2_ atmosphere, which exhibited an exceptionally catalytic activity and high selectivity as well as excellent reusability. It was interestingly found that as high as 75.6% yield of glycolic acid over 2.3 wt% Bi/β was achieved from cellulose at 180 °C for 16 h, which was superior to previously reported catalysts. Experimental results combined with characterization revealed that the synergetic effect between oxidation active sites from Bi species and surface acidity on H-β together with appropriate total surface acidity significantly facilitated the chemoselectivity towards the production of glycolic acid in the direct, one-pot conversion of cellulose. This study will shed light on rationally designing Bi-based heterogeneous catalysts for sustainably generating glycolic acid from renewable biomass resources in the future.

## 1. Introduction

The limited availability of non-renewable fossil resources together with ever-increasingly stringent legislation and mandates on global carbon neutrality strategies have motivated the extensive search for alternative renewable resources to resolve wide-ranging social, economic, environmental, and political issues that need to be addressed today to ensure a sustainable future for all [1,2,3]. Biomass is the only promising alternative carbon resource in nature and possesses tremendous potential and economic feasibility for replacing fossil resources to renewably generate bio-based liquid fuels, high-value chemicals, and materials [4,5,6]. Cellulose, the largest component of renewable biomass resources that are non-edible by humans and inexpensive, thus has aroused considerable attention and is regarded as one of the most promising and great potential substitutes for producing liquid fuels and highly valuable chemicals [7,8,9,10,11]. Particularly, glycolic acid (GA) is a very important biomass-derived high added-value platform with a hydroxyl and a carboxyl group, and exhibits the properties of both alcohol and organic acid, which has been widely applied in, but not limited to, metal cleaning, skin-care agent formulation, and industrial rusted removal together with food processing, especially in polymer degradation materials and pharmaceutical engineering materials [12,13,14,15,16]. GA represents a high and strong market demand, which was estimated to 310.4 million USD in 2020 and projected to reach approximately 531.5 million USD with an annual growth rate of 8.0% by 2027 [17].

In the past few years, great progress on direct hydrolysis and selective oxidation cleavage of C–C bonds of cellulose to GA has been made, yet the yield of GA was less than satisfactory, which substantially hindered the practical application of GA in industry. For instance, Jin et al. applied CuO as catalyst for transforming cellulose into GA under alkaline conditions at 573 K for 300 s, giving only a GA yield of 14.9% together with acetic acid (AA) yield of 5.2%, formic acid (FA) yield of 7.1%, and lactic acid (LA) yield of 13.8% [18]. Later, Han et al. reported that Keggin H_3_PMo_12_O_40_ could directly convert cellulose into GA in aqueous solution, which achieved a GA yield of 49.3% at 180 °C for 1 h and in 0.6 MPa O_2_ atmosphere [13]. Afterwards, similar catalysts of Dawson H_4_SiMo and H_3_PMo were reported, which afforded a maximum GA yield of 47.5% at 170 °C; this was mainly because heteropolyacids not only have high Brønsted acidity, but also have pre-eminent oxidation potency during the reaction [19]. However, these catalysts encountered some bottlenecks, including complicated recyclability, energy-consuming in separation, and industrial application limitations. Therefore, developing highly efficient heterogeneous catalysts that coordinate acidity and oxidation performance for realizing highly selective converting economically feasible cellulose resource to GA in an O_2_ atmosphere will be highly desirable but enormously challenging.

It is well known that zeolite H-β as a solid acid catalyst or metal support is potentially attractive due to its inherent properties, three-dimensional interconnected channel structure of rings containing 12 oxygen atoms, good thermal stability, and strong surface acidity, and has been widely applied in catalytic conversion of biomass resources into biofuels and highly valuable chemicals, especially for controlling hydrolysis of cellulose to glucose because of its well-defined crystal structure, unique shape selectivity, and high thermal stability [20,21,22]. Therefore, H-β zeolite has great potential and is promising for hydrolysis of cellulose and many kinds of catalytic reactions.

Previously, numerous heterogeneous catalysts with oxidation function have been recognized by inserting transition metal, such as Mo, V, Fe, Co, and Mn, into the framework of molecular sieves in highly dispersed forms [23,24,25,26,27]. However, these metal elements have the disadvantages of high cost and toxicity, so it is difficult to apply them on a large scale in industry. Non-noble metal Bi element, which is abundant on the Earth, and is not only less toxic, but more importantly, it can also shuttle between oxidation states, is emerging as an active and cost-effective catalyst with prospects for industrialization [28,29,30]. For example, Sun et al. reported that Bi/Bi_2_O_3_ showed high photocatalytic activity for the reforming of biomass-derived feedstocks [31]. In addition, Bi_2_O_3_ can be used as an efficient cocatalyst for selective photoelectrocatalytic (PEC) glycerol oxidation to DHA, achieving the PEC oxidation of glycerol to DHA with a high selectivity of 75.4% [32]. Inspired by this knowledge, we hypothesized that a rationally designed carrier would be crucial to incorporate Bi species as oxidation active sites into H-β to construct a multifunctional catalyst for hydrolysis and oxidation of cellulose to GA. To our knowledge, there are sparse reports on the catalytic performance of Bi-decorated β zeolite catalysts in the conversion of biomass resources into liquid biofuels and highly valuable chemicals. Therefore, it is highly desirable to develop more efficient and stable Bi-based heterogeneous catalysts for the production of GA starting from cellulose in aqueous solution and O_2_ atmosphere.

In this work, we reported a cost-effective and facile strategy for constructing Bi-decorated β zeolite catalysts that can selectively produce GA from cellulose by one-pot method. The resultant catalysts were thoroughly characterized with respect to their structure and performance. Importantly, the as-prepared catalysts not only could facilitate hydrolysis of cellulose to intermediate glucose in the presence of acidic condition in the first step, but also provided remarkable oxidation property that promoted the retro-aldol of intermediates by selective cleavage of C–C bonds and further oxidation into GA, exhibiting much higher catalytic activity than the ever previously reported catalysts. This study provides a promising potential and lays a good foundation for future research on promoting the transformation of renewable biomass resources into GA in industry.

## 2. Materials and Method

### 2.1. Chemicals

Microcrystalline cellulose (average particle size of 20 µm, polymerization degree of 250) was purchased from Sigma-Aldrich Co., LLC (St. Louis, MO, USA). Formic acid (FA, 99%), 1,3-dihydroxyacetone (DHA) (98%), glyceraldehyde (Gly, 99%), and acetic acid (AA, 98%) were commercially obtained from the Sinopharm Chemical Reagent Co., Ltd. (Shanghai, China). Zeolite H-β (Si/Al = 40) was purchased from the Nankai University Catalyst Plant (Tianjin, China). 5-hydroxymethylfurfural (HMF, 98%), erythrose (98%), glucose (99%), fructose (98%), and Bi(NO_3_)_3_·5H_2_O (99%) were purchased from Shanghai Macklin Biochemical Co., Ltd (Shanghai, China). Glycolic acid (GA) (98%), lactic acid (LA) (98%), levulinic acid (LeA) (98%), and glyceric acid (GlyA) (98%) were bought from Shanghai Aladdin Bio-Chem Technology Co., Ltd (Shanghai, China). Other reagents were all in analytical grade and used without further purification.

### 2.2. Catalyst Preparation

The Bi-decorated β catalysts were prepared by grafting Bi species onto H-β zeolites using bismuth nitrate as precursor. A known amount of Bi(NO_3_)_3_·5H_2_O (0.0474 g, 0.0718 g, 0.0967 g, 0.1222 g) was dissolved in 20 mL ethylene glycol under ultrasonication for 30 min, then 1.0 g H-β zeolite powder was slowly added and magnetically stirred at 80 °C for 8 h. The resulting mixture was filtered and washed repeatedly with deionized water until neutral filtration. Thereafter, the as-prepared Bi/β catalyst was evaporated under vacuum at 60 °C overnight and finally calcined in a muffle furnace at 550 °C for 6 h. The Bi/β catalysts with different Bi loading (1.26 wt %, 2.30 wt %, 3.40 wt %, 3.78 wt %) were identified as Bi/β-1, Bi/β-2, Bi/β-3, and Bi/β-4, respectively. The actual content of Bi was determined by inductively coupled plasma emission spectrometry (ICP-MS). The synthetic method of other metals (Mn, Cu, Co) decorated H-β and metal loading were the same as that of Bi/β-2 mentioned above.

### 2.3. Catalyst Characterization

X-ray diffraction (XRD) were determined using an X-ray diffractometer (Smartlab-3 KW, Rigaku, Japan) using Cu Kα as the source of radiation. The morphology was analyzed by field emission scanning electron microscope (SEM, S4800 instrument, HITACHI). The textural properties of catalysts were measured by N_2_ adsorption at −196 °C using a Micromeritics ASAP 2460 M nitrogen adsorption analyzer (BET). Before the nitrogen adsorption, samples were outgassed at 300 °C for 4 h. Raman analysis was carried out using a HORIBA Raman spectrometer and the spectra were obtained with the green line of an argon-ion laser (785 nm) in a micro-Raman configuration (Raman). X-ray photo electron spectroscopy (XPS) was performed on an AXIS Ultra DLD spectrometer (Shimazu, Japan) with Mg Kα radiation as the excitation source. Diffuse reflectance UV-Vis spectra of samples were recorded at wavelengths ranged from 200 to 800 nm on a UV-3600 spectrophotometer (UV-Vis, Shimazu, Japan). The surface acidity of catalysts was characterized by using NH_3_ temperature-programmed desorption (NH_3_-TPD) conducted on an AutoChem II 2920 chemisorption apparatus. The Bi/β catalysts were pretreated at 300 °C with a heating rate of 10 °C/min in flowing helium for 1 h until saturation. After cooling to 110 °C, catalysts were saturated in 10% ammonia diluted with helium for 2 h, then the samples were purged with helium flow (25 cm^3^/min) for 2 h. Finally, the samples were heated at a constant rate of 10 °C/min to 650 °C with helium flow. The acidity type and strength of catalysts were recorded by pyridine adsorption infrared spectroscopy (PyIR) on a Perki-nElmer-FT-IR Spectrometer. The Bi/β catalysts were first degassed at 300 °C and a vacuum of 6.0 × 10^−3^ Pa for 1 h. Then, the catalysts were exposed to pyridine vapor for 30 min. Subsequently, the catalysts were heated to 150 °C under a vacuum of 6.0 × 10^−3^ Pa.

### 2.4. Catalyst Experiments

In a typical reaction procedure, catalytic conversion experiments were performed in a 25 mL stainless steel autoclave equipped with a magnetic stirrer. A total of 0.1 g microcrystalline cellulose and 80 mg catalyst along with 12 mL deionized water was added into the autoclave, which was tightly sealed and then flushed with N_2_ three times to remove the ambient air, and then was pressurized to 2 MPa with O_2_ at room temperature. Subsequently, the reactor was heated to the desired reaction temperature and kept for desired reaction time with a stirring rate of 600 rpm. After completion of the reaction, the reactor was quickly cooled to room temperature with an ice-water bath. All the experiments were replicated three times, and the mean values were reported.

### 2.5. Product Analysis

The products were quantitatively determined by high-performance liquid chromatography (HPLC, Shimadzu LC-16) equipped with RID and a Bio-Rad Aminex HPX-87H Ion Exclusion Column; moreover, 5 mM H_2_SO_4_ was used as the mobile phase (0.5 mL min^−1^ of flow rate). The column temperature was set at 50 °C. The conversion of cellulose and the yields of products were calculated using external calibration curves derived from standard solutions as follows:

Cellulose conversion (%):(1)$$X=\left(1-\frac{\mathrm{mass}\;\mathrm{of}\;\mathrm{unreacted}\;\mathrm{cellulose}}{\mathrm{mass}\;\mathrm{of}\;\mathrm{initial}\;\mathrm{cellulose}}\right)\times 100$$

Yield of product *i* (C%):(2)$$X_{i}=\left(\frac{\mathrm{moles}\;\mathrm{of}\;\mathrm{carbon}\;\mathrm{in}\;\mathrm{product}\;i}{\mathrm{moles}\;\mathrm{of}\;\mathrm{carbon}\;\mathrm{in}\;\mathrm{initial}\;\mathrm{cellulose}}\right)\times 100$$

## 3. Results and Discussion

### 3.1. Characterization of Catalysts

The structure and morphology of H-β and Bi/β with different loading are shown in Figure 1. It was found that all catalysts had typical topological characteristics of BEA type structures with two diffraction peaks at 2θ = 7.5 and 22.5°, confirming that the structure of H-β zeolite was not seriously destroyed after incorporation of Bi [33,34,35]. Nevertheless, as the loading of Bi was increased, a significant decrease in the intensity of characteristic diffraction peaks was observed, indicating a partial loss of crystallinity. It was worth mentioning that no apparent diffraction peaks of Bi_2_O_3_ clusters were observed on Bi/β, indicating that Bi species were highly dispersed on the surface of pristine H-β or incorporated in the zeolite framework, thereby potentially improving the intimate contact between Bi species and H-β [29,33]. The SEM images revealed that all catalysts showed similar morphologies to that of pristine H-β without bulk bismuth oxide species on the surface of Bi/β catalysts, which further demonstrated that the structure of H-β zeolite remained intact (Figure 2).

The N_2_ adsorption–desorption isotherms and the pore size distribution of H-β and Bi/β catalysts are shown in Figure 3, and the structural properties and loading of Bi detected by ICP-MS are displayed in Table 1. As described in Figure 3a, it was noted that all catalysts showed the mixed isotherms of I and IV with H4-shaped hysteresis loops, indicating that both micropores and mesopores existed in the catalysts, which were further confirmed by the pore size distribution curves displayed in Figure 3b [36]. In addition, it could be clearly observed that the hysteresis loop decreased with the increase of Bi loading, indicating that the number of internal holes of catalysts decreased. Moreover, it can be found that the BET surface areas slightly decreased from 556.62 to 509.59 m^2^/g, and pore volume decreased obviously from 0.575 to 0.394 cm^3^/g with increasing Bi loading, which may be ascribed to the Bi species that entered into the inter-crystal channels of H-β, resulting in channel blockage or the occupation of exchange sites, as well as pore openings in the β zeolite matrix; it was well consistent with the XRD result [20].

In order to evaluate the Bi/β catalysts more comprehensively and subtly, the state of Bi species in the catalysts were investigated by UV-Vis DRS (Figure 4). Due to the transition of electrons from the oxygen valence band, the bulk Bi_2_O_3_ has a large absorption band at about 400 nm, whereas H-β showed a small absorption band and a large absorption band at around 215 and 274 nm, respectively [29]. Nevertheless, for Bi/β catalysts, the absorption band was blue shifted by about 50 nm, which may be ascribed to the interaction of isolated bismuth species with H-β support [30]. The increased band gap and blue shift tendency decreased with increasing Bi loading. The absence of peak at 400 nm in Bi/β catalysts indicated that there was no Bi_2_O_3_ species on the surface of Bi/β, indicating bismuth species entered the framework of Bi/β [32]. Similarly, the Raman spectra also showed that there were no peaks corresponding to Bi_2_O_3_ appearing in the Bi/β catalysts, revealing Bi_2_O_3_ was highly dispersed within the matrix of H-β. These observations were consistent with the results from XRD, SEM, and UV-Vis DRS (Figure 5).

The surface acidity and strength of catalysts were measured by NH_3_-TPD technique. As illustrated in Figure 6, it could be observed that H-β showed an intense desorption peak at around 210 °C and a broad desorption peak at 300~600 °C, which was attributed to weak and strong acid sites, respectively. This result indicated that weak acid sites were predominant while along with fewer strong acid sites on the H-β [37]. After introduction of Bi, a redistribution of acid sites on the Bi/β were distinctly observed. Specifically, the intensity of desorption peak centered at around 210 °C conspicuously decreased, whereas the desorption peak at 300~600 °C slightly shifted toward a lower desorption temperature, and the desorption peak areas evidently decreased compared to the H-β. This phenomenon could be reasonably explained by the incorporation of Bi species that caused the emergence of a new bump peak at around 250~450 °C, demonstrating the existence of medium acid sites on the surface of Bi/β. Moreover, increased Bi loading from 1.26 to 3.78 wt %, and the desorption peak areas slightly increased, indicated that the total acidity of Bi/β catalysts increased gradually due to formation of more weak and medium acid sites derived from more Bi species. These results indicated that the incorporation of Bi could not only transform the strong acid sites into medium acid sites, but also adjust the acid amount and strength of catalysts.

To further identify the Brønsted acid sites (BA) and Lewis acid sites (LA) on the catalysts, the catalysts were recorded by PyIR (Figure 7). The two bands at 1544 cm^−1^ and 1450 cm^−1^ were presented on all catalysts, which were attributed to the Brønsted and Lewis acid sites, respectively. In addition, it could be discerned that another band at 1490 cm^−1^ was ascribed to the combination of Brønsted and Lewis acid sites [38,39,40]. Moreover, as shown in Table 2, the amount of Lewis and Brønsted acid sites gradually increased with increasing Bi loading from 1.26 to 2.3 wt %, which was largely attributed to the presence of medium acid sites derived from the increase of Bi species. However, further increasing loading of Bi caused the amount of Lewis and Brønsted acid sites as well as the total acidity to decrease slightly, which was probably due to the coverage of Bi oxides active sites, whereas the amount of Brønsted acid sites decreased pronouncedly compared to the Lewis acid sites. It was mainly because the Bi^3+^ cations partially replaced the protons of the framework, which resulted in a decrease in the intensity of the Brønsted acid sites and the original acidity in the structure [22]. However, the relative ratio of the Brønsted to Lewis acid sites in the Bi/β catalysts was initially increased from 0.82 to 0.96 and then decreased to 0.88, demonstrating that the acidity and strength of Bi/β catalysts can be achieved by adjusting loading of Bi, and the Bi/β catalyst has also been proven to be capable as a potential bifunctional catalyst with Lewis and Brønsted acid sites.

The chemical environment of Bi element that was in interaction with support H-β was subjected to XPS measurement (Figure 8). It was apparent that the two binding energies were at 158.05 and 163.35 eV of pure Bi_2_O_3_, corresponding to Bi 4f_7/2_ and Bi 4f_5/2_, respectively. However, in the case of Bi/β catalysts, the binding energies of Bi 4f_7/2_ and Bi 4f_5/2_ were slightly higher than the corresponding values of Bi in bulk Bi_2_O_3_ [29]. This shift towards higher binding energies may be due to the decrease in electron density around Bi which resulted from the strong interaction between Bi species and H-β support (Figure 8a) [32]. The O 1s XPS spectra of Bi/β can be divided into two peaks at 531.2 and 532.9 eV, which can be assigned to lattice oxygen (O*_Latt_*) and surface adsorbed oxygen (O*_ads_*), respectively [23]. It is noted that the relative peak intensity of chemical adsorbed oxygen was clearly higher than that of lattice oxygen because of their higher mobility (Figure 8b).

### 3.2. Catalytic Conversion of Cellulose to GA over Various Catalysts

Table 3 displays the experimental results for hydrolysis and oxidation of cellulose over various catalysts at 180 °C for 10 h in a 2 MPa O_2_ atmosphere. It was found that the control experiment without any catalyst gave a GA yield of 7.7%, AA yield of 7.6%, negligible LA yield of 1.5%, and glucose yield of 1.0%, accompanied by minor amount of DHA (2.6%), FA (2.0%), traces of GlyA (0.3%), and 5-hydroxymethyl-2-furancarboxylic acid (HMFCA) (0.2%); this phenomenon was probably due to the H^+^ produced from auto-catalysis reaction of water at high temperature conditions, indicating that it was difficult to convert cellulose into GA under purely hydrothermal conditions (Table 3, entry 1) [7]. Furthermore, sole H-β afforded a 14.9% GA yield with a quite limited cellulose conversion of 40.8%, whereas it was obviously higher than that of no catalyst, and concomitantly, many kinds of by-products, including glucose (11.0%), LA (3.7%), FA (0.6%), AA (2.5%), GlyA (1.3%), and HMFCA (6.0%), were detected, indicating that H-β was favorable for hydrolysis of cellulose to glucose and further formation of GA due to its inherent Lewis and Brønsted acid sites (Table 3, entry 2). Remarkably, incorporation of 2.3 wt% Bi loading into H-β conspicuously improved the catalytic activity with cellulose conversion of 80.3% and GA yield of 51.9%, almost 4-fold greater than that of pure H-β. Additionally, only 6.7% LA yield was produced accompanying some other kinds of by-products such as AA (6.3%), FA (1.2%), glucose (6.9%), GlyA (0.6%), and DHA (1.2%), which was mainly because the Bi/β have the respective advantage of single component in catalysts; this result proved the significant synergetic effect between H-β and Bi species could control the conversion of cellulose toward formation of GA, and simultaneously suppress the formation of by-products LA, GlyA, and HMFCA (Table 3, entry 4). Additionally, it was worth noting that the reaction solution was colorless, revealing that Bi/β efficiently promoted the [2 + 4] retro-aldol of glucose from hydrolysis of cellulose as well as its further oxidation to GA rather than [3 + 3] retro-aldol of fructose to LA or dehydration of fructose to HMF [19]. To further clarify the effect of Bi active sites on the catalytic activity for the reaction, the control experiment using pure Bi_2_O_3_ as catalyst gave a GA yield of 21.8% that was a higher value than the blank reaction, further suggesting the indispensable role of Bi_2_O_3_ for conversion of cellulose to GA (Table 3, entry 10). Intentionally, physically mixing H-β (equimolar β to Bi/β-2) with Bi_2_O_3_ (equimolar Bi to Bi/β-2) resulted in a remarkable increase in GA yield of 42.6%, which showed a much higher yield toward GA relative to that of either constitute, reflecting that there was an obvious synergistic effect between H-β and Bi_2_O_3_ during the reaction process (Table 3, entry 11). However, the catalytic activity of physically mixed catalyst fell far below that of Bi/β-2 catalyst, probably ascribed to the strengthened interaction between Bi species and H-β. Moreover, Bi/β-2 possessed mostly weak acid sites together with few amounts of medium acid sites as well as higher ratio of Brønsted to Lewis acid sites. Based on these results, it was speculated that the synergetic interaction between H-β and Bi_2_O_3_ was considered to be the most important factors affecting the catalytic activity of Bi/β-2 catalyst in conversion of cellulose to GA.

To further elucidate the catalytic activities of different metal elements, Co/β, Cu/β, and Mn/β with the same loading equivalent to the Bi/β-2 were further studied under identical conditions. It was surprising to note that Bi/β exhibited an extraordinarily high catalytic activity and selectivity towards GA, accompanied with similar by-products distributions. Their catalytic activities were as follows in the order of Bi-β > Cu-β > Co-β > Mn-β, revealing that different metal elements had a perceptible effect on the catalyst activity for the reaction, presumably from the differences between their oxidation ability and pore structures (Table 3, entry 7–9). Moreover, the role of Bi active sites on the catalytic performance was further investigated. It was clearly observed that the strength of Lewis and Brønsted acid sites, total acidity, as well as the ratio of B/L increased gradually with increase of Bi loading from 1.26 to 2.3 wt %, in a good accordance with the yield of GA, suggesting that Bi played a crucial role in the cleavage of C–O bonds and C2–C3 bonds in catalytic conversion of cellulose. Nevertheless, further increase of Bi loading to 3.78 wt % resulted in subtle reduction in total acidity and the yield of GA from 51.9 to 34.1% (Table 3, entry 3–6). In contrast, the yields of undesirable by-products were increased, which was mainly because introduction of excessive Bi species blocked the internal holes inside Bi-β catalysts, resulting in the coverage of available active sites, and a decrease in BET surface areas together with strength of Lewis and Brønsted acid sites [37]. Therefore, it was rational to infer that the significantly improved catalytic activity was mainly due to simultaneously containing both Brønsted acid sites from surface hydroxy groups and oxidation active sites from Bi species partially isolated within Bi/β catalysts, which generated a synergistic effect for hydrolysis and selective oxidation reaction of cellulose. Moreover, it was inferred that hydrolysis of cellulose to glucose occurred in the first step during the whole reaction; Bi species were responsible for [2 + 4] retro-aldol and the formation of GA, and a moderate oxidative activity was indispensable for the selective oxidation of the resultant intermediates into GA. The catalytic activity is dependent not only on the total surface acidity and acid strength but also on the oxidation active sites on the Bi/β, as testified by the results of NH_3_-TPD and PyIR [30].

The reaction temperature is one of the most important factors on the catalytic activity for conversion of cellulose. As depicted in Figure 9, the yields of all the products were negligible for the optimum Bi/β-2 catalyst at 120 °C, indicating that the catalytic reaction proceeded to a very low degree owing to the inter- and intra-molecular hydrogen bonds of cellulose molecules. As the reaction temperature was then increased from 120 to 180 °C, the conversion of cellulose was raised rapidly from 30.4 to 80.3%, and the yield of GA increased remarkably from 0.2 to 51.9%, respectively. Meanwhile, the yields of other by-products and reaction intermediates, such as LA, AA, FA, DHA, AcA, fructose, glucose, and LeA, were also gradually increased, which was mainly ascribed to the fact that the reaction rate was accelerated at an elevated temperature that speeded mass transfer. Nevertheless, further increasing reaction temperature up to 200 °C would result in an almost completed cellulose conversion with GA yield of 37.5%. Meanwhile, the yield of LA evidently increased, whereas no humins was observed, but the yields of both AA and FA increased continuously. Moreover, the yields of DHA, GlyA, LeA, and AcA disappeared slowly at 200 °C, which was probably because higher temperature provided high energy and meanwhile accelerated the side reactions, such as decomposition of GA into AA and FA [18,41,42]. This phenomenon indicates that the complete transformation of highly stable and water-insoluble cellulose requires a higher reaction temperature that can facilitate the hydrolysis and fragmentation of cellulose.

The time-course profile showed that the conversion of cellulose was increased gradually from 52.3% to almost completely converted over Bi/β-2 catalyst from 4 to 16 h, and the corresponding yield of GA was continuously increased from 23.6 to 75.6%, suggesting a longer reaction time was favorable for the cleavage of C–O and C–C bonds in the conversion of cellulose and formation of GA (Figure 10). The maximum yield of GA with 75.6% was obtained at 180 °C for 16 h, which was noticeably higher than those of ever previously reported heteropolyacid catalysts [13]. A large amount of glucose was detected after a shorter reaction time, indicating glucose was the reaction intermediate. Moreover, it was observed that the yield of AA was raised continuously from 1.5 to 7.6% and then decreased slowly. Meanwhile, the yield of FA showed an increasing tendency from 0.3 to 2.4%, probably due to the decomposition of GA or LA to low carbon products. Moreover, a little amount of AcA yield was observed because of dehydration of LA [8,43]. However, further prolonging reaction time to 18 h resulted in a sudden reduction in the yield of GA of 25.2% due to its decomposition to FA and other by-product, such as CO_2_ [44]. Similarly, it was noted that the yield of DHA was raised initially from 0.1 to 1.2%, and finally disappeared at 18 h, indicating too long reaction time will cause a decrease in the yield of GA, mainly owing to the further decomposition of GA.

The relationship between catalyst dosage and the yields of products was further investigated (Figure 11). Evidently, as catalyst dosage was increased from 0.04 to 0.08 g, more available active sites would be afforded, which were beneficial for conversion of cellulose and gave a remarkably increasing yield of GA from 24.5 to 51.9%. However, further increase in the catalyst dosage from 0.08 to 0.12 g would afford excessive active sites to cellulose and intermediates, and they simultaneously accelerated the parallel reactions, such as dehydration of fructose to HMF and further rehydration of HMF to FA and LeA in acidic condition, or aldehyde-alcohol condensation as well as further composition of GA, thereby reducing the yield of GA [45]. Moreover, the agglomeration of catalyst particles resulting from high catalyst dosage also caused the decrease in catalytic activity, indicating an appropriate amount of catalyst could promote the conversion of cellulose to production of GA.

Subsequently, the effect of the initial cellulose amount on the yields of products was investigated at 180 °C for 10 h. As presented in Figure 12, it could be seen that when the amount of initial cellulose was increased from 0.05 g to 0.25 g, the yield of GA firstly increased and then gradually decreased, which may be attributed to the fact that relatively high initial cellulose amount led to polymerization of cellulose molecules, which resulted in coverage of the active sites of catalyst, and which resulted in insufficient numbers of active sites for the substrate. It was found that the yield of glucose continuously decreased from 17.4 to 4.3%, whereas the yield of AA gradually increased from 5.7 to 17.2%. Nevertheless, the yields of GlyA, DHA, and AcA gradually increased.

To comprehensively describe the catalytic activity of Bi/β-2, a variety of reactants were intensively followed, including cellobiose, fructose, and glucose, as well as sucrose as starting substrates (Figure 13). We found that the yield of GA starting from glucose was significantly higher than that of starting from fructose. Given the remarkably higher yield of GA using glucose as reactant than that derived from cellulose, it was speculated that the breaking of C–C bonds was the crucial step in the process of cellulose conversion to GA. Moreover, 21.3 and 27.8% of GA yield were obtained using sucrose and cellobiose as substrates, respectively. Conspicuously, the yield of GA from cellulose was highest among the substrates tested. It was demonstrated that the Bi/β has a great potential for practical applications in the production of GA platform compounds from biomass feedstocks.

### 3.3. Reaction Mechanism

As for the catalytic hydrolysis and oxidation of cellulose into GA in an O_2_ atmosphere, this usually involves hydrolysis of cellulose into glucose, [2 + 4] retro-aldol condensation of glucose to glycolaldehyde, [2 + 2] retro-aldol of erythrose to C2 intermediate, followed by oxidation of glycolaldehyde to GA. Concurrently, partial glucose goes though isomerization to fructose, and retro-aldol condensation of fructose to DHA and glyceraldehyde, which can be converted into glycolaldehyde and formaldehyde, and ultimately are oxidized to GA, FA, and AA. Additionally, dehydration of fructose and rehydration of HMF to FA and LeA are also easy to occur during the reaction [13,16,19,46,47,48]. Therefore, it is a very complex and synchronous reaction process. Combined with experimental results, it was clearly observed that intermediates glucose and fructose were almost fully converted over Bi/β-2 catalysts at 180 °C for 10 h, and the yield of GA starting from glucose was significantly higher than that of starting from fructose, testifying that the primary reaction route was the initial hydrolysis of cellulose into glucose and [2 + 4] retro-aldol reactions of glucose to GA rather than [3 + 3] retro-aldol and dehydration reaction of fructose to LA [18,49]. In addition, negligible HMF and LeA were detected, revealing the dehydration of fructose and rehydration of HMF occurred as a result of Brønsted acid sites. Particularly, a very small amount of HMFCA was formed, which was mainly ascribed to oxidation of HMF [50]. It was interesting to note that no erythrose and glycolaldehyde were observed in the products, which was likely because of their instability at high temperature condition, which went through the [2 + 2] retro-aldol and was quickly oxidized into GA, which was well consistent with the previous reports [19,39,49]. Moreover, trace amount of AA was observed due to quick cracking of glyceraldehyde or DHA. Therefore, the conversion of cellulose to GA involved the depolymerization or hydrolysis of cellulose into glucose in the presence of Brønsted acidity, retro-aldol reactions of glucose to produce glycolaldehyde and erythrose, and then glycolaldehyde further oxidized into GA [41,51,52]. In previous reports, the heteropolyacid catalyst gave a GA yield of about 49.3% together with an HMF yield of 15% for the reaction. Nevertheless, in this study, the Bi/β produced primary product GA and negligible HMF, showing a better catalytic selectivity towards GA because of reinforcing [2 + 4] retro-aldol condensation and appropriate surface acidity. Based on the above results and the previous literature, the cascade catalytic conversion of cellulose contains depolymerization of cellulose, the isomerization of glucose, and oxidation of intermediate, which can be synergistically catalyzed by the Lewis and Brønsted acid sites as well as oxidation active sites on the Bi/β-2 catalyst in one-pot reaction; the proposed reaction mechanisms are shown in Figure 1.

### 3.4. The Reusability of Catalysts

The stability of catalyst is very important for potential industrial applications. Figure 14 displays the results for the reusability of Bi/β-2 catalyst in the cellulose hydrolysis and oxidation at 180 °C for 10 h. There was a pre-eminent stability without an obvious decrease in catalytic activity after three successive cycles. The spent catalysts were recovered by calcination at 550 °C for 6 h for the next run. As shown in Figure 14, the yield of GA slightly decreased from 51.9 to 49% after the first cycle, and decreased to 47.5% after two cycles, and it was mainly due to the leaching of Bi active sites during the reaction process. Notably, the ICP-MS analysis from reaction solution after circulation proved that negligible Bi was lost during the reaction, and the content of leached Bi was lower than 0.14% in all three cycles. The XRD results of the regenerated Bi/β catalyst confirmed that the structure and morphology of catalysts were still well maintained without notable changes after three consecutive cycles, but the intensities of the characteristic diffraction peaks were slightly weaker, implying a slight decrease in crystallinity (Figure 15). These results suggested the good reusability, stability, and potential catalytic capability for future applications in the catalytic conversion of biomass to GA.

## 4. Conclusions

In summary, we developed a series of Bi-decorated zeolite β catalysts and achieved highly selective conversion of cellulose to GA in an O_2_ atmosphere. This study provided a novel approach by adjusting the Bi loading to control the C2–C3 cleavage of cellulose for the formation of GA. We found that with the 2.3 wt% Bi loaded, it was more favorable for production of GA from cellulose hydrolysis and oxidation, due to its appropriate total acidity, high ratio of B/L acid sites, and excellent oxidation activity. It is proved that it can synergistically catalyze the hydrolysis of cellulose and the selective oxidation of aldehyde groups to GA, achieving a GA yield of 75.6% at 180 °C for 16 h. Importantly, the as-synthesized catalyst could be used repeatedly for multiple times. The insights given in this work might provide instructive clues for the development of Bi-based heterogeneous catalysts for selective transformation of biomass feedstocks to GA and derivatives.

## Data Availability

Raw data of this article are available upon request to corresponding author.

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
