# Peer review of "Bismuth-Decorated Beta Zeolites Catalysts for Highly Selective Catalytic Oxidation of Cellulose to Biomass-Derived Glycolic Acid"

_ijerph, 2022, doi:10.3390/ijerph192316298_

Round 1
Reviewer 1 Report
Catalytic conversion of cellulose to liquid fuel and high-value platform chemicals still represents a challenging issue for cellulose conversion. This work has reported a bismuth-decorated β zeolite catalysts (Bi/β) for highly efficient hydrolysis and selective oxidation of cellulose to biomass-derived glycolic acid in an O2 atmosphere. The catalyst reported had high catalytic activity and selectivity as well as good reusability. There have several questions for the authors before the probable publication.
(1) The marks to indict author’s affiliation can not match their displays, and a obvious larger font size of “180 oC” can be found in the Abstract. These typos should be avoided.
(2) Line 116: “A known amount” is not clear.
(3) Line 119: “the sample” should be clarified. Other same questions in the article should be corrected.
(4) Line 159: For the product analysis, how to quantify the product by using HPLC? Detailed calculation should be introduced, for example, what authentic sample were used. A wondering question is can various products be effectively separated by HPLC and detected only by RID? The raw HPLC chromatography is suggested to provide.
(5) Statistical evaluation is missing., so it is difficult to evaluate the significance of the data presented in this paper. Please provide standard deviation or error bars in the table and figure.
(6) 75.6% yield of glycolic acid was achieved by using commercial microcrystalline cellulose in this research. Line 19-21, “which was superior over previously reported catalysts”. As we know, the property of raw materials is critical for catalytic efficiency. Are the raw materials used same or similar when you did the comparison? How about the effect of other types of cellulose?
Reviewer 2 Report
Manuscript Number: ijerph-2062284
title.
Bismuth-decorated beta zeolites catalysts for highly selective 2 catalytic oxidation of cellulose to biomass-derived glycolic acid
The article reviews the effect of Bi on the conversion of cellulose to glycolic acid. Authors report a high conversion, after a very long operating period (16h) at 180 °C in oxidizing conditions.
Recommendation:
Accepted with minor corrections
Comments to the authors.
Major comments
In TPD section. What about the used mass of the samples? The data is important for the right interpretation of figure 6.
Page 12. Line 370
Do the authors refer to the mass transfer coefficient?
Page 13 L 411
How was the agglomeration detected? It occurs always?
Minor comments
Pag 4, 144
The acidity type and strengthen of sites?, support? catalysts?
Author Response
Dear Reviewer,
Thank you very much for the excellent and professional review on our manuscript (Manuscript ID: ijerph-2062284). According to your suggestions, we have revised our manuscript and the corresponding responses are listed as follows:
(1) In TPD section. What about the used mass of the samples? The data is important for the right interpretation of figure 6.
Response: We thank the reviewer for pointing it out. The mass of sample determined by NH3-TPD technique was 50 mg.
(2) Page 12. Line 370
Do the authors refer to the mass transfer coefficient?
Response: Thank you very much for your valuable comment. We only studied the effect of reaction temperature and stirring rate on the reaction efficiency, and it was found that appropriate stirring rate was 600 rpm.
(3) Page 13 L 411
How was the agglomeration detected? It occurs always?
Response: Thank you very much for your good suggestion. A small amount of catalyst usually can not cause the occurrence of agglomeration, only when the dosage of catalyst was very superfluous.
(4) Page 4, 144
The acidity type and strengthen of sites? support? catalysts?
Response: We thank the reviewer for pointing it out. The acidity type and strengthen was about that of catalyst, and we have revised it in our Revised Manuscript.
Round 2
Reviewer 1 Report
The authors have appropriately addressed my questions and I would like to recommend an acceptance of this paper.